# Survivin Interference and SurVaxM as an Adjunct Therapy for Glioblastoma Multiforme

**DOI:** 10.3390/cells14100755

**Published:** 2025-05-21

**Authors:** Willie James Elliott, Nandini Gurramkonda, Maheedhara R. Guda, Andrew J. Tsung, Kiran K. Velpula

**Affiliations:** 1Department of Cancer Biology and Pharmacology, University of Illinois College of Medicine Peoria, Peoria, IL 61605, USA; wjellio2@uic.edu (W.J.E.); nandiniguramonda17@gmail.com (N.G.); gmreddy@uic.edu (M.R.G.); andrew.j.tsung@ini.org (A.J.T.); 2Department of Neurosurgery, University of Illinois College of Medicine Peoria, Peoria, IL 61605, USA; 3Illinois Neurological Institute, Peoria, IL 61605, USA; 4Department of Pediatrics, University of Illinois College of Medicine Peoria, Peoria, IL 61605, USA

**Keywords:** SurVaxM, surviving, Baculoviral IAP repeat containing 5 (BIRC5), glioblastoma (GBM)

## Abstract

Glioblastoma, IDH wild-type WHO Grade IV, is a devastating diagnosis in pediatric and adult populations with a poor prognosis and median overall survival of less than two years. Despite the advent of the Stupp protocol and advances in neurosurgical tumor resection techniques, there has been minimal change to both the quantity and quality of life in individuals diagnosed. Provided the extensive research on survivin’s association with glioblastoma tumor microenvironment, this review suggests that priming the individual’s immune systems to the tumor-promoting protein may reduce tumor burden through multiple mechanisms, including the arrest of the G2/M phase, microtubule dysfunction, induction of autophagy, and ultimately activation of apoptosis in glioblastoma cells. SurVaxM, a multiple peptide, survivin-specific vaccine, may assist in tumor cell destruction by eliciting the production of cytotoxic T-cells specific to survivin-expression glioblastoma tumors. Although phase I and II clinical trials suggest relatively safe adverse effects and potential efficacy, additional research is necessary to evaluate further how this vaccine may compare to standard treatment.

## 1. Introduction

### 1.1. Overview of GBM and Current Treatment Modalities

Grade IV astrocytoma, historically referenced as glioblastoma multiforme (GBM), is the most common adult primary central nervous system malignancy, with a median overall survival reported between 10 and 16 months [1,2,3,4]. As of 2021, the WHO classification of tumors of the central nervous system has delineated glioblastoma through isocitrate dehydrogenase (IDH) expression, with IDH wild-type high-grade gliomas being glioblastomas and IDH-mutant high-grade gliomas being grade IV astrocytoma or oligodendrogliomas [1,5]. These devastating tumors are often discovered in middle-aged to older individuals through imaging studies (e.g., MRI), illustrating a supratentorial lesion typically located in the frontal or temporal lobes [6,7]. Individuals suspected to have GBM will undergo maximal safe tumor resection and subsequent molecular and histopathological tumor characterization to optimize treatment. Shortly after surgical resection, temozolomide and radiotherapy are incorporated into the patient management strategy to augment tumor growth and prolong survival. This trio of surgical management, chemotherapeutics, and radiation, termed the Stupp regimen, has been the gold standard of GBM treatment for decades, notably improving median overall survival from 10.7 to 15.3 months [8]. Proper optimization of patient treatment modalities warrants a thorough investigation of the tumor molecular environment. The status of IDH activity alone can drastically affect prognostication in GBM-bearing patients [9,10]. The complex molecular interplay in the glioblastoma microenvironment is crucial to understanding the diagnostic process and how tumor genotype can influence patient management.

Prior to recent updates on the classification of central nervous system tumors, glioblastoma was separated into two categories based on having an IDH mutation or expressing the wild-type gene. In 2021, the World Health Organization solely defined glioblastoma as an IDH wild-type WHO grade IV astrocytoma, while the IDH-mutant counterpart is designated as a WHO Grade IV astrocytoma or oligodendroglioma [5]. Isocitrate dehydrogenase is a metabolic enzyme found in glioma and non-nervous system tumors that has five isotypes, with IDH-1 and IDH-2 comprising most mutations in glioblastoma [11]. This citric cycle enzyme primarily mediates the conversion of isocitrate to alpha-ketoglutarate, utilizing NADP+ as a cofactor. Mutations to IDH are believed to disrupt the glioblastoma tumor environment by reducing the antioxidative effects of NADPH and disrupting gene expression. Diagnostic testing for the IDH-1/2 mutations often utilizes a combination of DNA sequencing and immunohistochemistry to detect the R132H missense mutation [12,13]. Although the use of DNA sequencing and immunohistochemistry is quite sensitive and specific in detecting the IDH1-R132H mutation, both techniques can take several weeks to receive results. In an effort to improve efficiency, researchers have utilized RT-PCR and CRISPR technology to improve diagnostic turnaround time while providing potentially superior sensitivity and specificity for detecting the mutation [14].

While IDH status is crucial in determining prognosis and proper diagnosis, O6-methylguanine-DNA-methyltransferase (MGMT) methylation status distinctively aids in predicting the response to temozolomide and radiation therapy. MGMT is an enzyme whose gene resides on chromosome 10, coding for a protein that primarily functions as a DNA-repair enzyme at the O6 residue of guanine [15]. The methylation status of the enzyme, corresponding to attenuated activity, determines the inability of GBM tumor cells to repair cytotoxic alkylation by temozolomide, ultimately yielding GBM cell death [16]. Individuals diagnosed with GBM that are IDH-WT- and MGMT-methylated, therefore, have a significantly better prognosis due to increased sensitivity to temozolomide and subsequent radiotherapy [17]. The CeTeG/NOA-09 phase III clinical trial noted that individuals with MGMT-methylated GBM tumors may respond significantly better to a combination of temozolomide and lomustine compared to the use of temozolomide alone [18]. Patients having MGMT-unmethylated tumors are more resistant to the effects of temozolomide. Nonetheless, results of the EORTC-NCIC phase III trial observe enhanced survival in patients with both methylated and unmethylated MGMT promoters but do not show a significant change in progression-free survival amongst the unmethylated MGMT promoter patients [19]. Treatment with additional chemotherapeutic agents, as suggested by the GLARIUS trial, may be beneficial in improving 6-month progression-free survival, although a significant difference in overall survival is not observed [20]. Due to poor response from temozolomide and radiotherapy in GBM-unmethylated patients, this population is suggested to pursue clinical trial therapy.

In addition to testing for IDH and MGMT-methylation status, other molecular markers should be tested to rule out the possibility of alternative glioma diagnoses. For IDH-WT WHO Grade IV astrocytoma (i.e., glioblastoma), histopathological detection of microvascular proliferation or necrosis is sufficient to diagnose [21]. In instances where these histological features are absent, further molecular analysis for a positive H3 G34 mutation or H3 K27 alteration will yield a respective diagnosis of diffuse hemispheric glioma WHO Grade IV or diffuse midline glioma WHO Grade IV, respectively. IDH-mutant gliomas, no longer classified as GBM, are often tested for 1p/19q codeletion, ATRX, TP53, and CDKN2A/B [22,23,24,25]. The presence of diagnostic markers, such as 1p/19q codeletion or the loss of expression of ATRX and TP53 tumor suppressors, aids in defining the diagnoses of oligodendroglioma or IDH-mutant astrocytoma, respectively. Nevertheless, despite researchers’ progress in glioblastoma diagnosis and management, median overall survival over the decades has increased minimally, and the quality of life of individuals after treatment is questionable at best [26,27]. Owing to the advances in understanding the glioblastoma tumor microenvironment, inhibitor of apoptosis protein (IAP) survivin has been proposed as an anticancer target for malignancies, including gliomas, for decades, and appears to be a considerable adjunct to GBM therapy [28].

### 1.2. Overview of Inhibitor of Apoptosis Protein Survivin

Survivin, specifically termed Baculoviral IAP repeat containing 5 (BIRC5), is a member of the inhibitor of apoptosis (IAP) proteins [28,29]. Identified in 1993, inhibitor of apoptosis proteins have since been well studied and found to play a crucial role in cell cycle progression and apoptosis [30,31]. Survivin is a relatively small 16.5 kDa protein comprising both a well-conserved N-terminal zinc-binding baculoviral IAP repeat (BIR) domain, assisting in mediating protein-protein interactions, linked to an amphipathic C-terminal alpha helix [31,32]. BIRC5 is believed to promote tumor cell replication, invasion, and metastasis through cell cycle dysregulation and abrogating cellular death. As depicted in Figure 1, survivin’s tumor-promoting mechanisms of cell cycle regulation and cellular dysfunction are more specifically related to complex and poorly understood protein-microtubule interactions that facilitate G2/M cell cycle transition, as well as inhibiting autophagy and apoptotic processes. These tumor-promoting processes that facilitate GBM growth may be downregulated by BIRC5 inhibitors such as YM155. BIRC5 is believed to attenuate apoptosis by closely interacting with microtubules in the G2/M phase of the cell cycle [33]. In particular, microtubule-based organelles, centrosomes, showcase dysregulation with the transfection of U251MG cells with survivin siRNA [34]. These microtubule-constructed organelles are critical in interphase and mitosis as they maintain cellular polarity and ensure proper chromosomal segregation [35]. In the setting of siRNA-mediated survivin downregulation, glioblastoma cells express centrosomal dysfunction through centrosome amplification and corresponding chromosomal instability [34]. Given its critical role in the interphase process, the disruption of the survivin-microtubule interaction in the cell cycle is associated with enhanced caspase-3 activity, corresponding with increased cell death during mitosis. This microtubule interaction may be facilitated by protein-protein interaction between survivin and Aurora B, a mitotic serine-threonine kinase that facilitates chromosomal condensation, spindle kinetochore attachment, and mediates alignment and segregation of sister chromatids during the transition of metaphase to anaphase [36,37]. These protein interactions throughout mitosis that facilitate the G2/M transition are evidenced by studies in multiple malignancies, demonstrating the significance of survivin overexpression in cell cycle maintenance and cellular replication [38,39].

The concept of oncolytic viral therapy began in 1912, with scientists beginning to use rodent models in the mid-twentieth century to contribute to novel viral propagation techniques in an attempt to produce tumor-destructive viruses [40]. Oncolytic viral therapy is an immunotherapy that can target tumor cells and spare healthy cells through viruses to destroy tumors expressing specific antigens and proteins [41]. SurVaxM works similarly to oncolytic viral therapy and is a multiple peptide vaccine conjugate that targets survivin, particularly abundant in Grade IV gliomas, through the activation of cytotoxic T-cells that mediate the destruction of GBM cells using direct cell-cell contact or cytokine release [42,43,44].

## 2. Survivin Cellular Localization Impacts Overall Survival

Provided the presumed role of survivin in assisting Aurora B kinase and microtubule organization in the G2/M transition, adequate cellular localization of the protein is critical in adequately mediating these interactions. Multivariate analysis appreciates a positive correlation between overall survival in glioblastoma patients and clinicopathological factors such as age, MGMT, survivin expression, and survivin localization [45]. Glioblastoma patients with nuclear expression of survivin were found to have a significantly shorter overall survival of 19.5 months in comparison to individuals with predominantly cytoplasmic expression (31.7 months). Other studies have identified that high-grade astrocytomas and gliomas expressing both nuclear and cytoplasmic survivin exhibited a poorer survival rate than individuals with tumors expressing only nuclear or cytoplasmic survivin [46]. Immunohistochemical analysis of fifty-one high-grade astrocytoma samples for cellular localization of survivin revealed a significantly shorter overall survival in patients with tissue samples testing positive for both cytoplasmic and nuclear expression than with survivin expression of either location alone [47]. It is sensible that nuclear predominance of survivin in glioblastoma tissues may be associated with poorer survival due to the nucleus-specific roles the protein undertakes during its assistance in the G2/M phase. It is as yet unclear why glioblastomas expressing both nuclear and cytoplasmic survivin have a poorer prognosis than individuals who have either a predominance of nuclear or cytoplasmic survivin alone.

## 3. Survivin Overexpression and Immunosurveillance in Glioblastoma

Multiple nervous system tumors illustrate a strong expression of survivin [48]. An aggregate of studies supports that survivin is particularly overexpressed in glioblastoma cells [48,49,50]. Retrospective analysis of various human astrocytic tumors revealed 80.6% survivin expression in primary GBM tumors and 100% in secondary tumors [51]. Additional studies show similar degrees of survivin expression in glioblastoma [52], with some depicting up to ninety percent of glioblastoma samples expressing survivin [53,54]. In glioma, higher-grade gliomas illustrate a higher survivin mRNA or protein expression than their lower-grade counterparts [48,55,56]. As BIRC5 expression becomes more predominant in higher-grade gliomas, there appears to be an opposing negative linear association with survival in animal and human models. Analysis of 144 frozen glioblastoma samples found that higher levels of mRNA expression in the tissues were inversely associated with lower overall survival of the respective patients [57]. Glioblastoma patients with tumors displaying negative survivin histology appear to have a longer overall survival than those with survivin overexpression [58]. Enhanced mortality in survivin-abundant tumors is likely related to the anti-apoptotic properties of the protein, thereby facilitating the continued replication of glioblastoma tumor cells. Although prognosis may favor patients with minimal survivin expression in their tumor, proper immunorecognition is critical in mediating survivin levels and regulating the tumor microenvironment. Adequate immunoregulation and surveillance may offer survival benefits in individuals with survivin-expressing glioblastoma. Immunohistochemical analysis of the peripheral blood of patients with gliomas noted 30% of samples to have IFN-gamma responses [54]. Glioblastoma-bearing patients who elicited T-cell responses and IFN-gamma to survivin peptide were also found to have a longer overall survival.

## 4. Survivin Regulation Impacts Glioma Tumor Viability

### 4.1. Survivin and Radioresistance in GBM

Survivin is attributed to radiation resistance in various central nervous system malignancies, including medulloblastoma and various forms of gliomas [59,60,61]. In the setting of ionizing radiation, the downregulation of the inhibitor of apoptosis protein by siRNA in T98 glioma cells yields significant cellular destruction, compared to a lack of apoptosis evident with ionizing radiation and intact BIRC5 protein [60]. Post-ionizing radiation, survivin inhibition by siRNA increased apoptosis and DNA strand breakage in high-grade glioma cells, further supporting a radioresistant property of the protein [62]. Additional methods of survivin attenuation in glioblastoma through the regulation of other molecular targets have been identified to impact radioresistance in glioma tumor models [63]. The expression of survivin has been found to correspondingly decrease with the use of MTOR inhibitor rapamycin. Reduction of survivin through rapamycin and subsequent radiation treatment yields a decrease in GBM cell viability, suggesting a radio-sensitizing effect of the cells in the absence of BIRC5. Although the mechanism of survivin downregulation by Rapamycin is unclear, direct survivin modulators have been identified with strong evidence supporting the radiosensitizing effects of the agents when used alone or coupled with standard treatment.

### 4.2. Survivin Inhibitor YM155 Enhances Radiosensitivity in Glioblastoma

YM155 is a small imidazolium-based compound that was first discovered to abrogate survivin promoter activity and induce apoptosis in hormone-resistant prostate cancer cells [64]. Since this initial study, YM155 has been well studied in multiple malignancies as an augmenter of BIRC5 [65,66,67], notably not displaying any significant inhibition of other IAP proteins in the family. Although first discovered in the context of prostate cancer, YM155 has since been analyzed in the setting of glioblastoma cell lines M059K and M059J, with an augmentation of cell growth reported at an IC50 of 30–35 nm [68,69]. In radioresistant and radiosensitive GBM cell lines, M059K and M059J, respectively, the application of survivin inhibitor YM155 yielded a 70% inhibition of cell viability, suggesting a radiosensitizing effect of YM155 on the radioresistant cell line M059K. The use of YM155 in other radiation-resistant GBM cell lines further displays concentration-dependent cytotoxic effects, apoptotic induction, and abrogation of survivin with a 30 nM concentration of the protein inhibitor [69]. Rapid cell division, a defining characteristic of glioblastoma cells, intriguingly maintains its rapid-dividing stem cell-like phenotype in the presence of irradiation [70], thus contributing to its radiation resistance and difficulty in clinical treatment. Treatment of GBM cells with YM155 before irradiation corresponded with a blockage of the tumor plasticity and dedifferentiation into a less severe phenotype. Provided the role of survivin with microtubule organization, particularly in the G2/M phase, inhibition of the protein by YM155 should produce a disorganization of microtubules within interphase and mitosis. As depicted in Figure 2, attenuation of survivin by the BIRC5 inhibitor is associated with destabilizing microtubules within the G2/M phase, thereby yielding downstream activation of caspase-mediated apoptosis within GBM cells. Treatment of U87 glioblastoma cells with the survivin inhibitor further results in a significant centrosomal overduplication, suggesting a role of survivin in interfering with the microtubule organization of centrosomes.

Although not explicitly stated in the literature, it is feasible to suggest that the radiation-resistant characteristics of survivin may be related to its anti-apoptotic properties in relation to microtubule organization in interphase and mitosis; this is evident by the disorganization of centrosomes under survivin inhibition and the corresponding decreases in cell viability appreciated. The rapid cell-dividing quality of glioblastoma cells further lends to its highly invasive nature; radio-resistant characteristics of survivin may indeed contribute to these malignant characteristics. Radio-sensitizing effects of YM155 orthotopic GBM xenografts have reduced cell viability and tissue invasion, corresponding with a reduction in tumor growth and prolonged survival in the murine models [71]. These results are further supported by survivin-depleted glioma cells that are noted to have a reduction in cellular proliferation and metastatic properties with the use of YM155 [72].

### 4.3. Survivin Downregulation Impacts Radiosensitivity Through Inducing Apoptosis and Regulating Interphase

Despite YM155 being one of the most well-studied survivin inhibitors for glioblastoma multiforme, inhibition of the IAP protein may also be approached through inducing survivin mutations, siRNA interference, and additional pharmacotherapeutic agents, as highlighted in Table 1. Missense mutations induced in survivin genes through amino acid changes have been associated with both a downregulation of survivin and an attenuation of cell growth, likely mediated by an induction of caspase-dependent apoptosis [73]. Indirect regulation of survivin gene transcription through the inhibition of associated transcription factor Tetra-O-methyl Nordihydroguiaretic acid (M4N) in GBM cell lines, correspondingly down-regulates survivin and multiple survivin splice variants [74]. The downregulation of the IAP protein and its splice variants is expectedly associated with an induction of apoptosis, a decrease in cell mitotic index, and an arrest of G2/M cell cycle progression. Small interfering RNA molecules have further been utilized to suppress survivin activity, with similar results of abrogating tumor viability [75]. The use of other survivin RNA downregulatory agents, including small hairpin RNA (shRNA), has shown considerable efficacy in assisting other inducers of apoptosis in promoting murine GBM cell apoptosis, cellular invasion, and angiogenesis [76]. Nanoparticles have been implemented to assist in the delivery of anti-survivin agents across the blood-brain barrier (BBB) with the promise of inducing apoptosis within glioblastoma cells and improving radiosensitivity [77]. Nanoparticles packed with miRNA plasmids against survivin displayed efficacy in reducing survivin expression and enhancing caspase 3/7 activity compared to the control PEG-SPA miRNA plasmid. Radio-sensitizing effects of the anti-survivin nanoparticles were illustrated through a significant increase in caspase 3/7 activity with varying dosages of irradiation in U87MG and LN229 GBM cell lines. Anti-survivin pharmacotherapeutics replicate similar results in inducing apoptosis and regulating the G2/M transition of glioblastoma cells. Cucurbitacin, a natural compound isolated from the fruits of the gourd family (e.g., squash), has been shown to increase the number of GBM cells in the G2/M phase and subsequently decrease the number of cells in G1 and S phase, suggesting cell cycle arrest and cell death [78,79]. Polarization of the cells toward the G2/M phase is believed to be related to an increase in GADD45γ, an isoform of the GAD45 family of proteins that assists in the G2/M cell cycle checkpoint for DNA damage [79,80]. Exposure of GBM cells to cucurbitacin also augments the proliferation of the malignant cells [78]. A closer analysis of the potential signaling pathways involved in reducing cell viability reveals the downregulation of EGF-induced FAK, AKT, and GSK3ß in glioblastoma cells. Provided the significance of epidermal growth factor (EGF) and focal adhesion kinase (FAK) in promoting tumor proliferation, migration, and angiogenesis of tumors, further investigation displayed a dose and time-dependent significant inhibition of proliferating GBM cells.

The plant-derived survivin inhibitor further regulates microtubule interactions throughout the cell cycle by aiding aurora kinase A and B [81]. Aurora kinases A and B act alongside survivin, inner centromere protein (INCENP), and Borealin/Dasra-B as a chromosomal passenger complex (CPC) that mediates mitotic centromere localization [82,83]. These proteins work closely together to facilitate proper centromere localization, thereby facilitating mitotic cell division. Reduced expression of even a single member of the protein complex can attenuate the expression of other complex proteins and induce mitotic derangements in cells [84,85]. The use of the plant-derived inhibitor cucurbitacin significantly depletes mitotic spindles and decreases cell viability in glioblastoma cells [81]. Although it is unclear whether the regulation of mitotic spindles was secondary to the abrogation of survivin or Aurora kinase enzymes, which both share a close interplay in microtubule organization in the cell cycle, these results nonetheless suggest the potential role of additional pharmacotherapeutic agents in disrupting GBM mitotic division. The synthetic production of survivin inhibitor AZTM, an azide-terminated survivin ligand derivative proposed by Abbott Laboratories, displays a significant cytotoxic effect on GBM cell lines, corresponding with cellular apoptotic characteristics [86]. Pharmacotherapeutics well studied in non-nervous system pathologies, including the anti-malarial agent Artesunate, intriguingly show promise for radio-sensitizing glioblastoma cells through possible downregulation of survivin by unknown mechanisms [87,88]. Cells exposed to the synthetic derivative of artemisinin and subsequently irradiated display a significant increase in caspase 3/7 activity and cell death. Molecular compound Parthenolide, a germacrene sequiterpene lactone with antitumor properties, also shows promise in downregulating survivin and inhibiting GBM cell viability through similar mechanisms of cell cycle arrest; interestingly, the compound was only capable of inducing apoptosis in minimal GBM cells [89,90].

**Table 1 cells-14-00755-t001:** Clinically relevant survivin inhibitors tested on various GBM cell lineages, along with their respective molecular structure, half-maximal inhibitory concentration (IC50), and pathophysiologic effects on glioblastoma tissues.

Survivin Inhibitor	Molecular Structure	GBM Cell Viability IC50	Observed Effects on Glioblastoma Tissue
YM155	Imidazolium compound	30–35 nM	Apoptosis induction, centrosomal duplication, radiosensitization
Cucurbitacin	Oxidized tetracyclic triterpenoid [91]	2.5 µM	Mitotic spindle depletion, G2/M cell cycle arrest through increasing GAD45*γ*, EGF-induced FAK, AKT, and GSK3ß cell inhibition
AZTM	Azide-terminated survivin ligand derivative	0.78–4.5 nM [92]	Apoptosis induction, radiosensitization
Parthenolide	Germacrane sequiterpenenoid	16 µM	Mild apoptosis induction, autophagy induction, G2/M cell cycle arrest
Piperine	Alkaloid derivative	120 µM	Cell migration reduction
Spironolactone	Antimineralicorticoid Steroid lactone	Unknown	Chemoresistance

### 4.4. Survivin Regulatory Agents Show Promise for Adjunctive Therapy with Standard Treatment

Although survivin gene downregulation and inhibitor agents demonstrate apoptosis-promoting and cell cycle regulatory effects favoring a negative tumor microenvironment, the efficacy of survivin inhibition in combination with the current standard Stupp protocol is critical in determining the clinical translation of such complex molecular interplay. Glioblastoma cells exposed to temozolomide, a first-line chemotherapeutic agent in treating GBM shortly after tumor resection and pathological analysis, and cationic gemini surfactants carrying anti-survivin siRNA, yielded a significant decrease in cell viability compared to the use of temozolomide alone [93]. MicroRNA-138 (miR-138), found to be downregulated in GBM, has been depicted as an inhibiting agent of BIRC5 when overexpressed [94]. Combination therapy with temozolomide and survivin augmentation through miR-138 increases the survival in GBM xenograft murine models compared with the use of temozolomide alone, suggesting a clinical benefit of survivin inhibition with standard chemotherapeutic agents for treating the malignancy. Pharmacotherapeutic agents aimed at survivin downregulation combined with agents of the Stupp protocol further support the benefit of utilizing adjunct therapy for glioblastoma. Piperine, an alkaloid derived from the plant-producing black pepper, has shown a considerable reduction in GBM cell migration when combined with temozolomide treatment [95,96]. Potassium-sparing diuretics such as spironolactone also appear to confer radiosensitizing benefits in multiple cancer cell lines, including glioblastoma, possibly through the attenuation of survivin [97]. Glioma stem cells exposed to spironolactone had a corresponding reduction in survivin, cell viability, and enhanced cell death. Spironolactone use in non-glioma cell lines further displayed efficacy in reducing chemoresistance when used with chemotherapeutics such as gemcitabine and Osimertinib, suggesting a possibility of enhancing radiosensitivity to GBM cells when used with temozolomide.

## 5. SurVaxM Vaccine Therapy as a Treatment Modality for Refractive Glioblastoma

SurVaxM is a peptide vaccine developed for glioblastoma multiforme tumor cells expressing the survivin protein [98]. The vaccine consists of a main synthetic survivin peptide conjugate, a long peptide with multiple cytotoxic T-cell epitopes, and a survivin-specific CD4+ T-cell to facilitate the recruitment of cytokines. Vaccine use in glioblastoma may induce an immunogenic response against BIRC5 and a subsequent reduction of survivin exosomes, possibly corresponding with enhanced GBM apoptosis and improved cell cycle regulation [99]. As of early 2025, phase I and II studies have been conducted to assess the relative safety and efficacy of the vaccine with gold-standard treatment. As depicted in Table 2, these clinical studies and future phase II studies are ultimately dedicated to identifying the efficacy of immune-mediated inhibition of survivin in glioblastoma malignancy. Immunologic inhibition of the survivin protein has demonstrated efficacy in reducing high-grade gliomas in murine models and depicting safety for use in its first phase I clinical trial [43]. Analysis of nine patients with survivin-positive gliomas was fairly tolerable among the sample, with few grade one and three effects noted. Two-thirds of the cohort reported localized erythema at the injection site, one-third noted fatigue, and two patients experienced myalgia. Hematological manifestations were additionally observed, with a third of patients expressing grade 1 lymphopenia and leukopenia. The majority of the patients further expressed an immunologic response to survivin, with most of the patients having a stable disease course over the subsequent year. Subsequent phase II studies of SurVaxM and adjuvant temozolomide in 64 patients who failed the Stupp protocol reveal an 11.4-month median progression-free survival, with an overall survival of approximately 26 months; the clinical benefit was apparent in methylated and non-methylated GBM patients [42]. The patients were further found to elicit an immune response to the vaccine, with the production of survivin-specific CD8+ T-cells and immunoglobulin G. Pediatric patients with glioblastoma may additionally mount an immunogenic response to survivin-expressing GBM tumors [100]. Children with glioblastoma or other high-grade gliomas who were treated with intramuscular injections of peptide epitopes for survivin, EphA2, and IL13Ra2, mounted T-cell-specific responses to the peptides displayed considerable tolerability throughout the participants. The potential efficacy of the SurVaxM vaccine for adjuvant GBM therapy is further supported by observed prognostic benefits noted in mice with survivin-focused oncolytic adenoviral virotherapy [101]. FDA-approved neural stem cells expressing CRAd-Survivin-pk7, an adenovirus displaying selectivity to gliomas through survivin promoter modification, correlated with increased survival in glioma-bearing mice. The adenoviral therapy further yielded enhanced cytotoxicity of the tumor cells when paired with standard GBM treatment of temozolomide and radiotherapy [102]. Intriguingly, GBM-bearing mice treated illustrated a dose-response relationship with respect to virotherapy exposure and median survival.

## 6. Conclusions

Survivin regulation in glioblastoma multiforme appears critical in maintaining a tumor microenvironment resistant or sensitive to therapeutic interventions. The downregulation of BIRC5 likely reduces tumor burden in glioblastoma-expressing mice and humans through microtubule destabilization in interphase and mitosis, thereby facilitating apoptosis in rapidly dividing cancer cells. In addition to the efficacy of tumor reduction noted through siRNA interference and the use of survivin-specific inhibitors, vaccine conjugate therapy dedicated to the inhibitor of apoptosis protein possesses considerable potential in priming the immune system against survivin and reducing the radioresistant qualities inherent to the malignancy. The advent of oncolytic adenoviral therapy and similar vaccines such as SurVaxM for the treatment of glioblastoma has been studied in phase I and II clinical trials, demonstrating a relatively safe adverse effect profile and comparable overall survival when used with temozolomide. Although survivin inhibition by SurVaxM may yield potential clinical benefit to respective patients, additional phase II and III trials are important in elucidating both the efficacy and benefit of the vaccine in comparison to gold standard treatments. To date, multiple institutions such as the University of California, San Francisco (UCSF) are working toward clinical trials, such as the SURVIVE trial, which compares the possible benefits of anti-survivin vaccine therapy to standard temozolomide treatment [104]. Additional phase II clinical trials are being conducted to assess how SurVaxM combination therapy with anti-PD1 agent pembrolizumab impacts overall and progression-free survival in glioblastoma patients [103].

## Figures and Tables

**Figure 1 cells-14-00755-f001:**
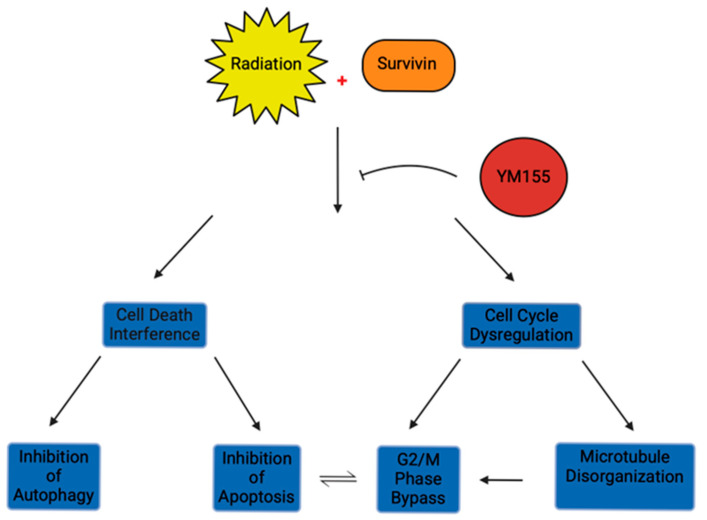
Illustrates the radioresistant mechanisms of the inhibitor of apoptosis protein, survivin, in the presence of radiation. Survivin primarily interferes with physiologic cell death and cell cycle dysregulation. Radioresistance in the context of cell death is dependent on BIRC5 downregulating apoptosis and autophagy. In contrast, cell cycle dysregulation results primarily from microtubule disorganization, thereby interfering with the G2/M checkpoint and resulting in apoptotic inhibition. BIRC5 inhibitor YM155 will reduce survivin production, thereby preventing cell death interference and cell cycle dysregulation and allowing for regulated apoptosis and cell cycle progression.

**Figure 2 cells-14-00755-f002:**
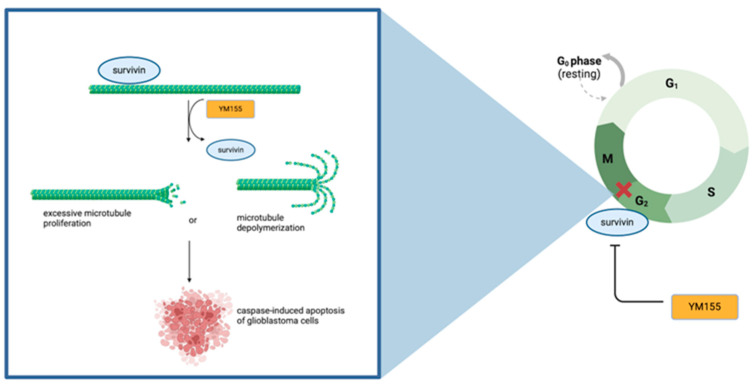
Depicts the blockage of the G2/M phase (depicted by the red X) by survivin with YM155 inhibition. Inhibition with the BIRC5 inhibitor (YM155) halts the G2/M phase bypass through survivin inhibition, resulting in the destabilization of microtubules. Survivin inhibition may destabilize microtubules through a hyperproliferating effect, corresponding to centrosome overduplication, or through microtubule depolymerization, associated with a decrease in mitotic spindle formation. Centrosomal overduplication and mitotic spindle depletion are associated with chromosomal abnormalities in mitosis and subsequent cell cycle arrest, promoting the apoptosis of glioblastoma cells.

**Table 2 cells-14-00755-t002:** Completed clinical trials and current trials, in addition to their respective clinical phase and notable discoveries, if applicable [103].

Trial Name/NCT Number	Clinical Phase	Status	Notable Discoveries
Phase I Study of Safety, Tolerability, and Immunologic Effects of a Survivin Peptide Mimic Vaccine (SurVaxM) in Patients with Recurrent Malignant Glioma	Phase I	Completed	Adverse effects elicited from patients include erythema (reddening) of the injection site, fatigue, myalgia, lymphopenia, leukopenia.
Phase IIa Study of SurVaxM Plus Adjuvant Temozolomide for Newly Diagnosed Glioblastoma	Phase II	Completed	No serious adverse effects elicited. Ninety-five percent of patients remained progression-free after diagnosis. Median progression-free survival (PFS) was 11.4 months, with an overall survival of 25.9 months. Survivin elicited a cytotoxic cellular and humoral response in vaccinated participants.
SurVaxM Plus Adjuvant Temozolomide for Newly Diagnosed Glioblastoma [104]	Phase II	In Progress	N/A
Phase II Study of Pembrolizumab Plus SurVaxM for Glioblastoma at First Recurrence	Phase II	In Progress	N/A

## Data Availability

The original data presented in the study are openly available through the National Library of Medicine on PubMed.

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
