# Peer review of "Survivin Interference and SurVaxM as an Adjunct Therapy for Glioblastoma Multiforme"

_cells, 2025, doi:10.3390/cells14100755_

Round 1

Reviewer 1 Report

Comments and Suggestions for Authors

The manuscript by Willie et al., is interesting; however major revision is required to be published in Cells journal.

Major comments

-The authors do not sufficiently describe the pathophysiology of GBM. They do not even mention TMZ, the most important treatment for GBM. Authors should modify the introduction section focusing on this. 

-Authors do not sufficiently describe the main genetic mutations involved in GBM (e.g. TP53, IDH, etc.). It will help the reader to better understand the purpose of the study and the importance of finding new therapies for GBM. 

-I suggest adding a figure about Survivin to better highlight its role. 

-The title "Superexpression of Survivin and Immunosurveillance in Glioblastoma" in paragraph 3 is the same as paragraph 4. Should they be in the same section? Please revise it.  

-Please check all abbreviations and typos in the manuscript. 

-Please exclude some older references and replace with an earlier one. 

Comments on the Quality of English Language

English could be improved to more clearly express the research

Author Response

  1. Comment: The authors do not sufficiently describe the pathophysiology of GBM. They do not even mention TMZ, the most important treatment for GBM. Authors should modify the introduction section focusing on this.

        Response: In the introduction, I provided a sufficient explanation of the role of temozolomide in the Stupp regimen on            page 1/16.

  1. Comment: Authors do not sufficiently describe the main genetic mutations involved in GBM (e.g., TP53, IDH, etc). It will help the reader to better understand the purpose of the study and the importance of finding new therapies for GBM.

         Response: In the introduction, glioblastoma genetic-driven diagnostic markers are described, with emphasis on major            diagnostic markers such as IDH mutation and MGMT methylation status; this information is on page 2/16. Other                       molecular markers that separate the diagnosis of GBM, Who Grade IV Astrocytoma, and oligodendrogliomas, are                     discussed on page 3/16; highlighting the following molecular markers: H3-G34, H3-K27, 1p/19q codeletion, ATRX, TP53,           and CDKN2a/b.

  1. Comment: I suggest adding a figure about survivin to better highlight its role

      Response: The role of survivin in regulating the G2/M phase and cellular apoptosis is not well understood as of current             literature. It would therefore be difficult to add a figure better illustrating its unclear role.

  1. Comment: The title “Superexpression of Survivin and Immunosurveillance in Glioblastoma” in paragraph 3 is the same as paragraph 4. Should they be in the same section? Please revise it.

        Response: We apologize for not discovering this error in our initial review of the paper. Paragraph 3 on page 5/16                     should  read, “Survivin Overexpression and Immunosurveillance in Glioblastoma,” and paragraph 4 should have, and                 now reads, “Survivin Regulation Impacts Glioma Tumor Viability.”

  1. Comment: Please check all abbreviations and typos in the manuscript.
    Response:
    The manuscript was reviewed for typos; all typos have been corrected. Abbreviations were also checked and are properly used throughout the paper.
  2. Comment: Please exclude some older references and replace with earlier ones.

       Response: Thank you for this feedback. Many of the references included in the 1990s are relevant to the discovery of               survivin and inhibitors of apoptosis proteins. Replacing these references would involve no longer including the historical            context of the protein, which we feel adds important information to the review.

Reviewer 2 Report

Comments and Suggestions for Authors

The review “Survivin Interference and SurVaxM as an Adjunct Therapy for Glioblastoma Multiforme” is a very well-articulated compilation of the available literature on the topic to date. It is well known that Glioblastoma is a devastating disease with poor prognosis and needs urgent attention for novel therapeutic strategies to combat it. Survivin is a promising target from the point of view of developing novel strategies and has been worked extensively. This review posits that priming an individual's immune systems to the tumor-promoting protein may reduce tumor burden through a variety of mechanisms in glioblastoma cells, given the extensive research on survivin's association with the tumor microenvironment of glioblastoma. 

The review is written in an easy-to-understand language and has assessed most of the available information in the field. I will highly recommend this for publication in this journal.

Author Response

  1. Comment: The review “Survivin Interference and SurVaxM as an Adjunct Therapy for Glioblastoma Multiforme” is a very well-articulated compilation of the available literature on the topic to date. It is well known that glioblastoma is a devastating disease with poor prognosis and needs urgent attention for novel therapeutic strategies to combat it. Survivin is a promising target from the point of view of developing novel strategies and has been worked extensively. The review posits that priming an individuals immune systems to the tumor-promoting protein may reduce tumor burden through a variety of mechanisms in glioblastoma cells, given the extensive research on survivin’s association with the tumor microenvironment of glioblastoma.

Response: Thank you for the great feedback and for recognizing the potential the review serves to the scientific community.

  1. Comment: The review is written in an easy-to-understand language and has assessed most of the available information in the field. I highly recommend this publication in this journal.
    Review:
    Thank you for expressing the benefits that this paper adds to the field of glioblastoma pharmacotherapy and the scientific literature.

Reviewer 3 Report

Comments and Suggestions for Authors
  1. They must discuss the role of Survivin in the chromosomal passenger complex (CPC) that is crucial to control mitosis and as direct caspase inhibitor
  2. Fig 2 is not clearly explained. The reference for these data must be included

3. Do the authors speculate  about some Glioma–specific  mechanisms that sensitize GB cells to Survivin inhibitors more than other cancer types 

Author Response

  1. Comment: They must discuss the role of Survivin in the chromosomal passenger complex (CPC) that is crucial to control mitosis and as a direct caspase inhibitor.

Response: Thank you for noting the significance of the CPC in regulating centromere localization with critical proteins such as survivin, aurora B kinase, and others. We made sure to include information on the crucial complex in section 4.3, page 8/16, in the second paragraph.

  1. Comment: Figure 2 is not clearly explained. The reference for these data must be included.

Response: Thank you for identifying the ambiguity in the explanation in Figure 2. We sought to better explain Figure 2 by clarifying that the G2/M phase is blocked by the YM155 survivin inhibitor. We further clarified that the subsequent centrosomal abnormalities are specifically associated with chromosomal abnormalities in mitosis, which ultimately result in cell death. The references for figure 2 have been clarified.

  1. Comment: Do the authors speculate about some Glioma-specific mechanisms that sensitize GB cells to survivin inhibitors more than other cancer types?

Response: There does not appear to be any glioma-specific mechanisms apparent in the literature that may make them more sensitive to survivin inhibitors than other cancer types. It is the susceptibility of other cancer types to survivin inhibitors that also makes GBM a consideration for such treatment.

Round 2

Reviewer 1 Report

Comments and Suggestions for Authors

The authors improved the manuscript according to reviewer suggestions.